# Public Housing and Household Savings—A Three-Decade Repeated Cross-Sectional Analysis

**DOI:** 10.3390/ijerph22081182

**Published:** 2025-07-28

**Authors:** Yi Zhang, Man Tsun Wong, Yik Wa Law, Paul Siu Fai Yip

**Affiliations:** 1National Engineering Laboratory for Big Data Analysis and Applications, Peking University, Beijing 100871, China; zhangyi03@pku.edu.cn; 2Department of Social Work and Social Administration, The University of Hong Kong, Hong Kong, China; wmtmatt@hku.hk (M.T.W.); flawhk@hku.hk (Y.W.L.)

**Keywords:** public rental housing, household savings, Hong Kong

## Abstract

Housing affordability is a major determinant of quality of life. Despite the relatively high GDP per capita in Hong Kong (HK) (USD 460,000), about one-third of the population lives in public rental housing (PRH) because they cannot afford private housing. Existing research estimating the benefits of PRH or direct housing supports faces methodological limitations. Addressing this research gap, our study adopts an “in-kind subsidy” approach to estimate the monetary value of PRH, quantifying how much less PRH households might save monthly if they resided in private rental units, after controlling for confounding factors. This paper examines the association of housing types and household savings by applying ordinary least squares (OLS) regression to compare savings among PRH tenants, non-PRH tenants, and mortgage-free homeowners, based on seven rounds of the Household Expenditure Survey data (1989/1990 to 2019/2020). PRH tenants saved significantly more than private housing tenants. In terms of household savings, the value of HK PRH has steadily increased from HKD 4483 in 1999/2000, to HKD 9187 in 2019/2020. For every dollar increase in income, a household would have the propensity to save 0.7 dollars in 2019/2020. Given limited public resources, our findings offer robust evidence regarding the value of public housing. The results underscore the importance of the equitable allocation and effective management of current PRH stock to enhance the upward mobility of low- to middle-income households amid limited housing resources in HK.

## 1. Introduction

Housing is a basic requirement for human survival, and thus it plays an important role in social stability. Nevertheless, housing affordability has become a major social and policy issue in many post-industrial countries and cities, including the United States, Germany, and certain leading metropolitan areas in urban China [1,2,3,4,5]. In such situations, tenants facing high rental costs may be forced to limit their use of basic necessities or reduce spending on other needs. It is not uncommon to see high rental burden in terms of rent-to-income ratios, such as 62.7% in the lowest income quintile in the United States [6], 35% in London, around 25% in Wales, Northern Ireland and England [7], 45% in certain economically leading urban cities in China [3], and over 30% in Australia [8]. To alleviate the housing affordability problem, governments often offer direct housing provision or similar public housing schemes to low-income households.

In the context of Hong Kong (HK), after World War II, HK experienced rapid urbanization and population growth, with housing shortages and affordability emerging as major obstacles to improving HK residents’ quality of life. This reflects the impact of limited housing supplies, scarce land for construction (only 7% of 1000 km^2^), high purchase prices, and high rents. The 2022 Annual Demographia International Housing Affordability Survey ranked HK as the least affordable housing market among 92 major markets in eight nations, with a record high price-to-income ratio of 23:2 in 2021 [9]. In HK, residents generally live in three types of housing: PRH, Home Ownership Scheme (HOS) housing, and private housing. Currently, about 858,000 households (approximately 28% of HK residents) are tenants in PRH flats [10]. Combined with the 16% of HK residents living in HOS units provided by the Government, nearly half of HK residents are directly supported by the Government’s housing policy, indicating a huge engagement of the Government in the HK housing market [11].

The HK Government’s active involvement in tackling housing issues started from an incident. A fire in the district Shek Kip Mei in 1953 left 53,000 residents homeless, prompting the HK Government to initiate its first public housing program to rehouse the victims of the fire in 1954 [12]. Thereafter, HK experienced a fast and persistent growth in population size, nearly 1 million for each decade until the 1990s [13]. The increase in population was largely due to inflows from mainland China, and the majority were not well-off [14]. The need to shelter such a rapidly increasing population triggered long-lasting large-scale construction of the PRH flats by the HK Government. As a consequence, the HK Government has been playing a determining role in handling housing issues thereafter. In addition to resettling the victims of the fire, the Government then launched a large-scale public housing program (PRH) in the 1970s, providing low-cost rental housing to eligible households. The Hong Kong Housing Authority (HA) was established in 1973 with an aim to support and implement the HK Government’s housing policy, such as the development of public housing estates. Currently, most PRH are managed by HA, while a few are managed by the Hong Kong Housing Society [15,16]. Public housing is generally rented at a lower-than-market price to meet the housing requirements of low-income urban dwellers. Apart from PRH, the Government also launched subsidized homeownership programs, such as HOS, which sells flats at discounted prices, usually around 30% below market price [17]. There are also other housing projects from the Hong Kong Housing Society and the Urban Renewal Authority.

Housing is not simply a commodity; it is an important pillar of stabilizing society. Given the extensive public housing market in HK, it is particularly important to ensure fairness in the distribution of government-led PRH. Addressing concerns regarding the equitable distribution of public housing estates involves not only economic considerations, but also the fairness of policies underpinning the housing security system.

The increasing queue of people waiting for public housing, as well as the widening gap in rents between private and public housing flats, has fueled controversy regarding the equitable and rational allocation of PRH flats. The average waiting time for general applicants of PRH is 5.7 years in 2024, far exceeding the targeted 3-year pledge [18]. Despite the Cash Allowance Trial Scheme that disburses cash allowance (ranging from HKD 1300 to HKD 3900 depending on the number of eligible household members) to PRH applicants waiting over 3 years [19], the low-income households eligible and queueing for PRH are in desperate need of direct housing aids instead of a cash allowance that could not even afford half of the normal rental prices.

To address such concerns, it is first necessary to have a good understanding of how much benefit PRH tenants actually obtain through the PRH program, particularly the value of a PRH unit as an in-kind subsidy, compared to non-PRH tenants.

### 1.1. What Are the Benefits of Public Rental Housing, and How to Quantify Them?

Researchers attempted to assess the benefits of similar public rental policies in other regions. Data Envelopment Analysis (DEA) is one method of efficiency evaluation that has been used in institutional efficiency evaluations. It measures the efficiency of a decision-making unit (which is the PRH program in this context) by evaluating its input/output ratio. For example, the DEA method has been employed to evaluate the efficiency of the public rental system in some mainland China cities [20]. The number of units built in PRH could be considered as the output; however, it would be nearly impossible to estimate the input from the Government on PRH programs due to the availability of data in HK, such as the labor costs, construction costs, and time costs relative to the number of PRH units. Researchers mostly take proxy measurements, for example, Liu and Pan [20] took the amount of government’s funds and investments as the input, but this is not an accurate measurement. DEA is not adopted in the present study in light of the unidentified input.

Another approach to evaluating the benefits of public housing is the cost–benefit approach. To quantify cost–benefit, the HK Government has adopted the imputed rent method (also known as rental equivalence or opportunity cost method) to convert the transfer-in-kind of the PRH into equivalent monetary value. This estimates the opportunity cost to the Government if a PRH unit is leased in a hypothetical open market [21,22]. This method has been adopted in some previous studies to estimate the cost and benefit of direct housing support or home ownership [23,24,25]. The difference between market rent imputation based on the average market rent of PRH buildings provided by the Rating and Valuation Department and the rent paid by the household is considered as the transfer-in-kind subsidy of the PRH. According to this method, the average benefits of the PRH units per month in 2006 and 2016 were HKD 240 and HKD 900, and further increased to HKD 1060 in 2021 [22]. However, studies have argued that figures obtained from imputation may not be accurate [26]. For example, applying an improved imputed rent method and a subsidized-to-market rent ratio method, Lui assumed a fixed housing subsidy for PRH tenants of HKD 3708 and HKD 1203, respectively [27]. Lopez and Yoshida [25] also pointed out that the major challenge of this approach is how to accurately capture the opportunity cost of the housing units. Hence, when considering housing tenure as a critical rental determinant, the imputed rent of PRH will not accurately capture the additional value of PRH. In summary, prior studies have shown that existing estimates for the benefits of public housing or direct housing supports face certain limitations and potential biases.

Facing this significant research gap, we take an alternative view of the monetary estimate of PRH flats, using an “in-kind subsidy” approach, that is, if a current PRH household were unable to obtain a PRH unit, how much less money this household might save each month if it lived in a private housing unit at the market rental price, after controlling for other factors.

### 1.2. Savings as Safety Net for Low-to-Middle Income Families

Low-to-middle income families seek wealth accumulation as a safety net against poverty, simultaneously strengthening their capacity and potential to move upward on the social ladder. Household savings, defined as the remaining balance of monthly disposable income after consumption expenditure, serve as a crucial indicator for a household’s financial situation and a security pillar against cash flow crises [28]. Savings provide families with a means of managing risks caused by unpredictable situations such as economic downturns, death, illness, debt, or asset mismanagement. Adequate savings can reduce a household’s financial vulnerability in the face of significant risk shocks [29,30].

Savings also provide a safety net during income fluctuations or emergencies, particularly for the elderly. Research highlights the vital role household savings play in maintaining living standards after retirement [31,32]. Quality of life for retirees largely depends on their financial security at retirement.

In addition, household savings have been increasingly important in an aging society. Insufficient savings would restrict elderly people’s capacity to face post-retirement financial challenges. The retirement protection system in HK follows the World Bank’s multi-pillar concept [33,34]. It consists of three pillars: government-provided programs, a mandatory employment-based provident fund, and private savings and familial support. It is uncertain whether the retirement protection from the HK Government is adequate, or will be adequate, considering the rapid population aging in HK. Zhu and Chou [35] noted a trend of an increasing proportion of elderly people financially relying only on themselves rather than on their adult children, simply because they have no children, which can be reflected in the persisting ultra-low birth rate in HK. In addition to the minimal income protection from the HK Government, the provident fund offers limited security for low-income retirees, as it only covers working people and the protection varies with income [36]. It is therefore a wise choice, if not the only one, to consolidate the private-savings-pillar to guarantee the quality of life after retirement, taking into account that human life expectancy continues to increase. Unfortunately, low levels of private retirement savings were found in HK workers [35]. Noticeably, savings are crucial for value-added investments, including financial capital, social capital, and personal endowment enhancement. They are particularly important for households with children or young adults, who require financial support for education, training, and social network expansion. However, low- to middle-income households often struggle to grow their savings, with housing costs contributing a large portion of household expenditure.

### 1.3. Housing and Savings

Besides serving as a safety net against future uncertainties, saving is often regarded as postponed consumption, which, from a life course perspective, serves concrete future life goals such as accumulating a down payment to buy a home [37]. Hence, in many ways, savings are fundamental for low- to middle-income families. Prior research has shown that in terms of both the absolute amount and the savings rate, low- to middle-income families tend to save much less than richer families—a pattern consistently observed in both developed and developing economies worldwide [38,39,40,41]. This could be a result of the burden of potential housing market losses, mortgage payments, or other forms of recurrent expenditure for richer families and homeowners [37,42,43], as saving rates may also fluctuate along with the housing prices or expected housing prices [44,45]. Housing characteristics, such as living area and household composition, were also found to have a critical effect on saving behaviors [46].

Alternatively, it has been observed that policy or societal events could affect saving behaviors. For instance, during the COVID-19 pandemic, emergency situations and relevant health protection policies, such as lockdown policies, had a significant effect on forced household saving, as this served as a means of coping with uncertainties and future financial challenges [47]. A housing policy reform in China in 1998 that abolished the provision or subsidization of public housing provided by employers, also substantially drove up household saving rates due to increased demand for private housing [45]. They also contended that the rising housing costs and underdeveloped mortgage market have further driven up the saving rates after the reform, as households see the urge and need to save up more for housing. Similarly, another study in China found that migrants’ saving rate significantly decreased after buying houses, falling below that of the local hukou population [48]. Previous studies have established that large-scale housing-related policies can have a profound impact on household saving rates. However, there remains a significant research gap in understanding the effect of public housing policies. The PRH scheme in HK is distinctive because it provides eligible households with highly subsidized rental units, granting them de facto lifelong security of tenure and the ability to transfer this right to their descendants. However, these households do not possess legal ownership of the property. This hybrid arrangement raises important questions about how such in-kind benefits influence household saving and consumption decisions, particularly compared to conventional models of homeownership or cash assistance.

Understanding the benefit of PRH as a type of in-kind subsidy is crucial for several reasons. First, it provides empirical evidence on how non-cash, non-ownership-based housing policies affect financial decision-making, household welfare, and intergenerational mobility. Second, with housing affordability and inequality becoming increasingly pressing issues globally, insights from HK’s experience can inform policy development in other high-density urban settings facing similar challenges. Third, the PRH scheme’s unique features—life-long security, transferability, and absence of legal ownership—allow us to isolate the effects of housing security from the wealth effects typically associated with ownership. This distinction is vital for understanding the true value and behavioral impact of in-kind housing subsidies.

By addressing this research gap, our study contributes to the broader literature on public housing, social welfare policy, and household economics. The findings can help policymakers better assess the trade-offs between different forms of housing assistance, and design more effective interventions to promote financial stability and social equity among urban populations.

This study used 30 years of population-wide expenditure survey data (1989/1990 to 2019/2020) carried out by the Census and Statistics Department (C&SD) of the HK Government, to test three research questions: (1) What were the over-time saving trends of and differences between three groups of households, namely PRH tenants, non-PRH tenants, and private homeowners with no mortgage? (2) How much less did non-PRH tenants save, compared with PRH tenants and private homeowners? and (3) What were the over-time changes in association between PRH and HK residents’ household savings?

## 2. Data and Methods

### 2.1. Data

Data from seven series of the HK Household Expenditure Survey (HES) data from 1989/1990 to 2019/2020 were analyzed. The HES is conducted once every five years by the C&SD. It collects up-to-date information on households’ consumption patterns with the purpose of updating expenditure weights of the Consumer Price Indices. The HES covers all domestic households in HK, except for those households receiving Comprehensive Social Security Assistance (CSSA), a means-tested financial subsidy provided to families in need.

The surveys adopt stratified proportionate sample design with geographical area and type of housing as the stratification factors. The sample sizes and response rates (in parentheses) were 4854 (71%), 5591 (76%), 6116 (80%), 6054 (81%), 5959 (77%), 6812 (72%), and 7056 (67%) households for the years 1989/1990, 1994/1995, 1999/2000, 2004/2005, 2009/2010, 2014/2015, and 2019/2020, respectively. Details of the sampling methods and profiles of the respondents have been presented in a series of C&SD publications [49,50,51,52,53,54].

### 2.2. Method

The HES collects information on the expenditure patterns of households by way of an expenditure diary. Each household member will need to record his/her daily expenditures in the diary, as well as providing information on regular payments and expenditures on infrequent purchases. Income covers earnings from employment, net income from self-employment, and alternative income sources such as interest, pensions, grants, and scholarships, while expenditure covers all actual purchases made during the survey period. Monthly household savings are computed as the difference between monthly household income and expenditure. The analysis included eligible households where there was at least one member in the household who was employed (an earner). Analysis excluded households with extreme (mostly negative) values in savings.

There were three analysis steps. Firstly, the proportion of households with no savings (monthly income equals or is smaller than monthly expenditure) and the average saving ratios was calculated. Saving ratios were calculated as the proportion of household savings relative to total income. Three series of average saving ratios were computed for all study period years: (1) unadjusted savings ratio, which included all households with negative savings; (2) adjusted savings ratio 1, which excluded all households with negative savings; and (3) adjusted savings ratio 2, which included all households with negative savings but recoded all negative savings values to zero.

Secondly, monthly household income was categorized into quintiles, and average monthly household savings were calculated within quintiles for each round of the HES data.

Thirdly, for each round of the HES data, ordinary least squares (“OLS”) regression models were calculated for the three housing subsets (PRH tenants, non-PRH tenants, and homeowners with no mortgage). In order to reduce the influence of extreme values on the estimates, the top and bottom 1% of the household savings data was deleted, before calculating each regression model. To ensure comparability of monetary values across the years, both income and savings were adjusted to the constant price for the final data collection period (2019/2020). Monthly household savings were regressed on household structure (including age and gender of household heads, household sizes, the number of residents aged 65 years and above, and the number of children younger than 15 years), household income, and accommodation type (PRH tenants as the reference group, non-PRH tenants, and homeowners with no mortgage). The regression model was as follows:$$\begin{array}{l} Savings=\beta_{0}+\beta_{1}\times Age+\beta_{2}\times Gender+\beta_{3}\times Household\;Size\\ \;+\beta_{4}\times No.Elderly+\beta_{5}\times No.Children+\beta_{6}\times Income+\beta_{7}\times Housing \end{array}$$

## 3. Results

An increasing proportion of households had positive savings over the study period. Figure 1 highlights that while 28.8% of households had negative savings in the year 1989/1999, this dropped to 14.7% by the year 2019/2020. The extra-large standard deviations (about 700–3200%) of the unadjusted savings ratio across the years indicated a substantial bias if calculations included all households with negative saving amounts. Thus, two series of adjusted saving ratios were reported, both of which indicated a persistently increasing trajectory over the study period.

Overall, savings increased as income escalated. This is shown in Table 1 (average savings by income quintiles across accommodation types). Such findings appeared to be generally consistent across time and accommodation type. There were exceptions however, such as among non-PRH tenants in the years 1989/1990 and 1994/1995, which might be explained by small sample sizes. Household savings seemed to be higher in the year 1999/2000 than in the neighboring survey rounds (years 1994/1995 and 2004/2005), potentially led by political turmoil and uncertainties upon the handover of sovereignty from Britain to China in 1997, as well as the downturn inproperty and rental prices starting from 1997 to 2003 [55]. Figures from the Rating and Valuation Department [55] show the rental indices in 1997 dropped from 134.5 to 100.0 in 1999 and further lowered to 73.6 in 2003. When the data from 1999 to 2000 is ignored, average savings increased over time among PRH tenants, although saving trends for non-PRH tenants and the homeowners were less clear. Noticeably, both the PRH tenants and the homeowners saved much more than the non-PRH tenants in each income group within the same data collection period. Such findings reflected significant redistributive effects on the income of the PRH units. There was no obvious pattern, however, in average savings comparing PRH tenants and homeowners.

Regression analyses confirmed the association of housing type and savings, which implies the potential impact of PRH flat tenancy as an in-kind subsidy on monthly household savings. Table 2 reports the findings of the OLS regression models. The overall model fit was more satisfactory for recent years than for earlier years in the study period. In the years 2009/2010 and 2014/2015, as much as 72% of the variance in savings could be explained by the explanatory variables. By contrast, the R-square value approximated 40% in 1989/1990, which suggested that the model had failed to take into account potentially important factors.

Table 2 also indicates that household structure, household income, and the types of accommodation were all statistically significant factors in predicting the average monthly household savings. The findings support the notion that household structure affects household savings. Females as household heads, larger household sizes, and a greater number of children were associated with fewer savings. On the other hand, greater numbers of elderly household members were associated with greater savings. The association between savings and the number of children was inconsistent however, as households with fewer children had fewer savings in 2014/2015.

The interpretation of non-significant results requires careful consideration, as it may be premature to dismiss potential associations. The insignificance may be simply due to small sample sizes. Since household structure influences both income and expenditure, it was included in the regression model for adjustment.

Household income was significantly and positively associated with household savings across all years in all models. Effect sizes varied little across the years, mainly staying between 0.5 and 0.6, and reaching 0.7 in the year 2019/2020. This suggests that all things being equal, for every dollar increase in income, a household would have the propensity to save at least 0.5 dollars across the data collection periods. This finding clearly suggests that increasing household income is the most direct and effective channel to improve household savings.

The type of accommodation was significantly associated with household savings. Compared with PRH tenants, non-PRH tenants saved much less, whereas in some years (1999/2000, 2004/2005, 2009/2010, and 2019/2020), homeowners saved marginally more. The coefficient for non-PRH tenants and homeowners is the amount of savings compared to PRH units. For example, the negative coefficient of −2302 in 1989/1990 for non-PRH tenants would mean an equivalent amount of reduced in-kind subsidy compared to PRH tenants. This indicated that for tenant households, whether living in PRH represented an in-kind subsidy of HKD 2302 in terms of household savings. As the increase in housing price had led to an increase in the rental cost as well, consequently, the estimated of the subsidy had increased from HKD 2302 in 1989/1990 to HKD 9187 in 2019/2020 for PRH tenants.

To further validate the robustness of our findings, we conducted additional analyses using a 20% sample drawn from the full dataset, due to data confidentiality and availability constraints imposed by the government. Specifically, we compared PRH tenants with non-PRH tenants by employing propensity score matching (PSM) based on household size, income, gender, age, and dependent ratio to minimize potential confounding effects. Both income and savings were adjusted to the constant price for the final data collection period (2014/2015). The sample sizes after PSM for 2009/2010 and 2014/2015 were 206 and 310, respectively, while the data prior to 2009/2010 was not used due to small sample size after PSM. Monthly household savings were estimated as the difference between household income and expenditure, and an ordinary least squares (OLS) regression was performed to quantify the impact of PRH as an in-kind subsidy on household savings.

The results are summarized in Table 3. These findings align closely with the main analysis, confirming that PRH as an in-kind subsidy significantly enhances household savings, with benefits reaching over thousands of HKD. The result also confirms that after PSM, for every dollar increase in income, a household would have the propensity to save at least 0.5 dollars. The consistency of the results across different samples and analytical methods underscores the robustness of the conclusion that PRH provides substantial economic benefits to tenants, effectively reducing their household expenditure and improving financial stability.

## 4. Limitations

There are important limitations in this study. First, the calculation of household expenditure may not have been comprehensive. Although it covered the major aspects of consumption, including food, housing, clothing, goods, transportation, and miscellaneous services, it did not equal the sum of all outflowing money per month, as the list of expenditure could be non-exhaustive. For example, it did not include gifts to charities, cash contributions/gifts to relatives and friends, and so on. For this reason, the gap between the monthly income and expenditure might not be the exact amount of household savings in real life scenarios. However, we believe the major aspects of consumption listed above are accurate captures of the average, most-seen consumption in our daily activities; therefore, we expect the gap in calculation would be rather marginal. With the legal authority of the C&SD and the way of collecting data with expenditure diaries, it is reasonable to presume that the expenditure recorded in this study is accurate.

Second, household income was pre-tax and hence did not reflect real disposable household income. The sample did not include households receiving income from government subsidies such as CSSA, hence a substantial number of households with low income, covering 319,200 recipients (approximately 4–5% of the total population, found mostly in the public housing sector), are not included in this study [56]. By this means, our study mostly compares the savings and in-kind subsidies of those who are not receiving a substantial amount of social security subsidies. Similar methods have been adopted in the literature and our findings of savings patterns are largely comparable with other studies [40,57,58,59].

Third, the price of any accommodation relies largely on its location and size [27]. Theoretically, these characteristics could affect the monetary value of PRH units. The present study refers to the monetary estimate of PRH flats as the net household savings, hence it is a limitation that we did not take into account the monetary value in terms of their location and the size of PRH. However, the size and design of PRH is standardized and regulated. PRH flats are located across almost all districts in HK (except Wan Chai District), and are provided in urban, extended urban, new territories, and island parts. Applicants would choose from one of the above four categories before queueing for the flats. PRH flats are also allocated with a priority to meet circumstantial needs, such as family size, age, and disability. Hence, we expect the differences in value of PRH in terms of housing characteristics would not be prominent, as the heterogeneity of PRH is limited.

It is suggested that further research could consider these differences in in-kind subsidy of PRH, although sample sizes in subsets of the HES data might not be sufficiently large to support such analyses.

## 5. Discussion

This paper presents new information on the potential impact of PRH tenancy on household savings in HK over the past 30 years. Addressing the research questions, three important findings were identified, which were consistent over the study period:(1)An increasing percentage of households were able to save money during the study period. Both the PRH tenants and the homeowners saved much more than the non-PRH tenants in each income group;(2)There were potential redistributive effects of PRH on household savings. After controlling for income level and household structure, PRH tenants saved much more than tenants in non-PRH flats, with the gap widening over the study period. An up-to-date figure of the value of PRH as in-kind subsidy was HKD 9186.9, when compared to non-PRH tenants;(3)Household savings were closely associated with household income, family demographic structures, and type of housing

### 5.1. Savings: A Novel Angle of Looking at PRH Value

Our study ascertains a novel and accurate way of capturing the value of PRH in terms of savings. By way of comparing tenants in terms of savings, it tackles the issue of having no ground for comparison between different households with divergent housing types.

The findings show that one PRH unit represents as much as over HKD 9000 household savings monthly on average for non-homeowners. This figure is income-controlled and household structure-controlled, indicating a substantial net value of PRH to its tenants in comparison with their non-PRH counterparts, regardless of their socio-economic status. This finding is critical, as it is indeed way beyond the imputed or estimated benefits in the prior literature; ranging mainly from HKD 240 to HKD 1060 per month in the Legislative Council Research Office [22], and up to HKD 3708 in Lui [27]. The discrepancies in prior, estimated benefits and the in-kind subsidy in our study indicate an improper audit and account of public resources in this aspect—as a hybrid of social resource and commodity, the value of public housing stock must be accurately accounted for, documented, and reviewed, to facilitate further policy review and evaluation. If policymakers underestimate the impact of PRH, the strength and efficiency of planning and implementing such policy may also be overlooked.

This study highlights the significance of PRH for low-income tenants in terms of opportunities for household savings and protection against future uncertainties. This is despite challenges such as long PRH waiting times. Our findings confirm that a housing policy that provides direct housing aid has a significant impact on household savings. There are other aspects of housing that can affect household savings, such as transport, communication networks, work, and lifestyle. Housing location will affect residents’ travel time and choice of transport. In Indonesia, the government provides support through housing improvement programs, direct supply of mixed-income housing by private developers, and cross subsidies. However, due to the lack of affordable housing in urban centers, low-income households are often unable to live there. Thus, they need to spend more time travelling, as well as more on transportation costs to engage in work and life activities. This will affect their household savings [60].

There are also additional policy factors that could have an impact on household savings. Several studies have reported significant effects of interest rates on household savings in different countries [61,62,63,64]. The affordability of housing will also affect household savings. For example, in addition to vigorously building government housing units and then selling them at low prices, the Singapore government has also provided strong financial support and introduced a series of incentive policies, including reducing interest rates, extending the repayment period and allowing the use of a Central Provident Fund for down payments. These housing policies provide a solid foundation for residents to achieve home ownership [65]. Future research which aims to empirically predict household savings could also take note of these attributes at a broader level.

### 5.2. Divergence Between PRH Tenants and Non-PRH Tenants

From the results, it is important to note that the value of PRH could be dynamic instead of stagnant. Over the time interval of 30 years, the benefit of PRH has been steadily increasing, especially outgrowing the non-PRH tenants. In the present social context, social resources have been consistently allocated to PRH tenants, as they are most often regarded as the group with high financial instability. Across the years, the relative value of PRH as a form of social protection has grown even more pronounced. This finding has important implications for other regions or countries facing similar affordability crises: large-scale PRH interventions can provide meaningful financial relief for low-income households, thereby promoting financial stability and potentially improving overall social welfare.

In comparison with PRH tenants, low- to middle-income non-PRH tenants have been facing the grimmest situation. For non-PRH households, the average household savings among the bottom income quintiles was negative, which means that many households must make a living through channels other than monthly income, most probably by using their previously accumulated household savings. Therefore, there is a pressing need for these households to reverse this situation. Such unsustainable savings patterns will undoubtedly put them at risk of deteriorating living standards or even falling into poverty, as studies found household savings, or even the availability of saving accounts, are effective protection against poverty or shocks in the short and long run [66,67]. Moreover, without sufficient savings, they are also at risk of being deprived of opportunities for personal development, social capital accumulation, and social upward mobility. In addition to providing public housing that serves as the baseline safety net for households in need, it is also necessary to formulate corresponding policies or schemes that target household savings and upward mobility for non-PRH low-income households.

### 5.3. Enhancing Circulation of PRH Stock

Beyond serving as evidence for the effectiveness of PRH, our findings also offer critical insights for the ongoing policy debates around the design and future direction of PRH schemes. In HK, where nearly half the population resides in public housing, yet affordability issues persist, there has been an ongoing discussion about how PRH schemes can be further optimized not just to provide shelter, but also to promote upward social mobility among low-income families. Historically, the life-long tenancy of PRH has been a cornerstone of HK’s housing policy. However, our results—showing the substantial economic advantage conferred by PRH—suggest that it may be timely for policymakers to reconsider whether such a tenancy is still the most effective way of allocating this valuable resource, or whether there are ways to better enhance the circulation of PRH.

Due to the previously underestimated benefits of PRH (often based on imputed rent methods), the government has lacked accurate measures of how much PRH tenants gain relative to those not allocated a PRH flat. This underestimation has contributed to disruptions in social cohesion and growing disparities between PRH and non-PRH households, prompting recent efforts by the government to address perceived imbalances, such as stricter eligibility checks for high-income PRH tenants. Researchers have suggested different ways of enhancing the circulation of PRH housing stocks. For example, Wong [68] has long been supporting a method of selling the PRH units to its current occupants at discounted prices, if they could afford and were willing to buy them. The idea of privatizing public housing has received criticisms for its inoperability and equity concerns. Forrest and Xian [69] argued that the privatization of public housing would mostly benefit the existing PRH tenants while leaving fewer housing options for the next generation. The strong evidence presented in this study provides a timely and concrete basis for a potential redesign of PRH tenancy policies. Our findings underscore the substantial benefits that PRH provides to low-income households, notably in terms of financial security and social protection. This strong benefit, while advantageous, can also act as a double-edged sword. On the one hand, it enhances social welfare by ensuring stable housing and reducing inequality; on the other hand, it may inadvertently diminish households’ incentives for upward mobility. When PRH is perceived as a lifelong safety net without clear pathways to ownership or income improvement, households might become complacent, leading to a stagnation in social and economic mobility. While there is no consensus on the feasible solution, policymakers should at least initiate a discussion on the currently stagnant and rigid PRH housing supply issue and review the policy tools on hand. The current review on “Well-off Tenants Policies” could be a starting point, but it is certainly not adequate. Specifically, PRH could be reconceptualized as a transitional platform that supports low-income households in building savings and progressing toward homeownership, rather than as a permanent destination, except for those who need PRH as a safety net.

Such a reform may not be the sole possible solution to enhance the circulation of public housing resources. For example, subsidized homeownership programs, such as the HOS, should be encouraged and put in place progressively. The largely discounted price of the HOS units makes them more affordable for the prospective homeowners from low- to middle-income families.

### 5.4. Steady Supply for PRH Units

Another avenue for reform lies in increasing the supply of PRH units. The Hong Kong Government has previously attempted to address the supply–demand imbalance, most notably through the Ten-Year Housing Plan, initiated by the first Chief Executive, Mr. Chee-Hwa Tung, in 1997, aiming for 70% homeownership by 2006. His policy, which projected a housing supply of about 85,000 units per year, faced backlash due to a 40% drop in private property prices during 1997 and 2003 [70,71]. It is arguable whether the decline in private property prices could be attributed to Tung’s housing policy, because its impact on the housing market was mixed with the Asian financial crisis in HK during the period 1997–1998, and SARS which prevailed in HK during the period 2002–2003. Li [72] concluded that the 85,000 policy has a good intention, but bad timing. As a matter of fact, the HK Government never achieved its goal of supplying 85,000 housing units per year, the annual average supply of housing was 60,800 units (38,900 public housing units and 21,900 private housing units) during Tung’s term, the period 1997–2002. The completion of new housing units only achieved the level of 85,000 in 2000–2001 [72]. The disruption of Tung’s housing policy was a reaction to the unstable housing market, but its abandonment under his successor, Mr. Donald Tsang, marked the non-responsiveness of the HK Government to timely monitoring an unbalanced development between supply and demand in the housing market. By contrast, there were only around 15,000 public housing units built per year during Tsang’s term. The public housing program was virtually abandoned during the period. Figure 2 shows the public housing production from 1998 to 2023. This resulted in a substantial shortage in housing supply compared to housing demand, which led to the significant increase in property and rental prices in HK. The evidence from our study not only affirms the considerable benefit of PRH but also highlights the risks of policy inertia and under-provision of public housing. Moving forward, policymakers should consider both increasing the supply of PRH and refining tenancy policies to ensure that PRH remains an effective tool for alleviating housing stress, improving household financial well-being, and fostering social mobility.

## 6. Concluding Remarks

This study provides robust empirical evidence on the association and changes over time in the association between housing type and savings rates, generating new insights on the potential impact of PRH in HK on household savings, based on seven rounds of Household Expenditure Survey data spanning three decades (1989/1990 to 2019/2020).

The findings reveal that PRH tenancies are significantly associated with higher household savings compared to tenants in private housing, even after controlling for income and household structure. Over time, the monetary value of PRH as an in-kind benefit has grown substantially, reaching over HKD 9000 per household per month in 2019/2020. This suggests that PRH plays a critical redistributive role in enhancing the financial resilience of low-income households.

There are growing signs that owning a property sets a clear boundary between higher and lower social classes. For most private housing tenants, there are no signs that their capability to purchase a home is increasing. The potential redistributive effect of PRH on household savings has widened over time, indicating a growing disadvantage for private housing tenants, particularly those in the low- to middle-income brackets. The consistently negative savings among the bottom income quintiles of non-PRH households are alarming and call for targeted policy intervention. These households are at greater risk of economic shocks, downward mobility, and intergenerational poverty. Instead of low-income families in PRH, the worst situation then appears to be for low- to middle-income tenants living in non-PRH flats, or still on the lengthy queue to PRH. It is critical for the HK Government to reassess its housing policies and seek more effective solutions in order to restore equilibrium in the housing market and improve living conditions for HK residents.

Housing affordability issues are not uncommon across the globe; however, this study provides robust empirical evidence as to how a housing intervention as large as such in HK could have a significant association with household savings, thereby affecting their ability to protect themselves against future uncertainties or to proceed with other life goals, including homeownership. The evidence also suggested possibilities for the HK Government to consider increasing the supply of PRH and enhancing the circulation of PRH by way of reconceptualizing the policy with a transitional nature. This study also underscores the urgent need for an updated and more accurate valuation of PRH in the government’s fiscal and social policy planning. We demonstrated the substantial protective effects on household savings of PRH units, as well as the widening savings gap between non-PRH households and PRH households. PRH has been a safety net for low-income earners in the past, but the rapidly growing waiting list for PRH in recent years highlights both the pressing demands for housing and the relative effectiveness of PRH in addressing affordability concerns. The value of PRH has been historically underestimated, and our findings suggest that a miscalculation of its actual benefits could lead to underinvestment in the public housing infrastructure. The HK Government needs not only to solve housing shortages and cost, but also to provide upward mobility for low-income earners. Given the substantial effect of PRH occupancy on household savings, and the tight supply of housing, our findings suggest an urgent need for an equitable and efficient allocation of PRH flats. We call for an enhancement of circulation in PRH stock, not solely through increasing supply, but also by implementing reforms that reinforce the fundamental purpose of PRH. Specifically, reforms should aim to uphold the value of PRH as a vital welfare resource and a means to promote upward mobility and homeownership. By aligning policies with these core objectives, we can ensure that PRH continues to serve as an effective tool for social equity and long-term sustainable development.

This study redefines the value of PRH not only as a shelter but as a mechanism for financial inclusion and social mobility. It reveals that housing policy is intrinsically linked to household economic behavior and thus should be viewed as part of a broader social protection strategy. Policymakers, non-profit agencies, and even financial institutions should recognize PRH tenants as having greater capacity for savings and consider ways to channel this into asset-building opportunities.

In conclusion, this study provides a timely and policy-relevant contribution to understanding the multifaceted role of public housing in HK. By quantifying its substantial impact on household savings, it calls for a paradigm shift: from viewing PRH merely as a commodity and welfare provision to recognizing it as a transition from poverty to financial empowerment, upward mobility and sustainable (self-sufficiency) development.

## Figures and Tables

**Figure 1 ijerph-22-01182-f001:**
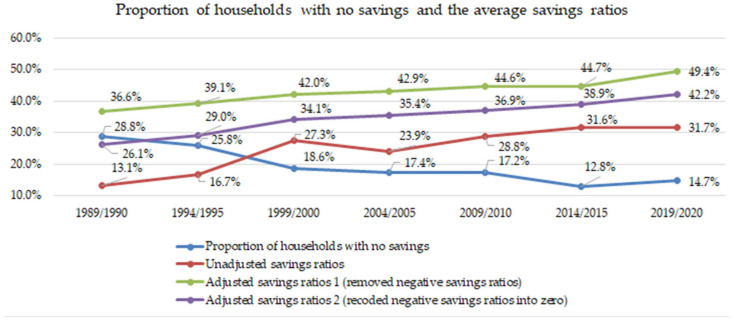
Proportion of households with no savings and the average saving ratios.

**Figure 2 ijerph-22-01182-f002:**
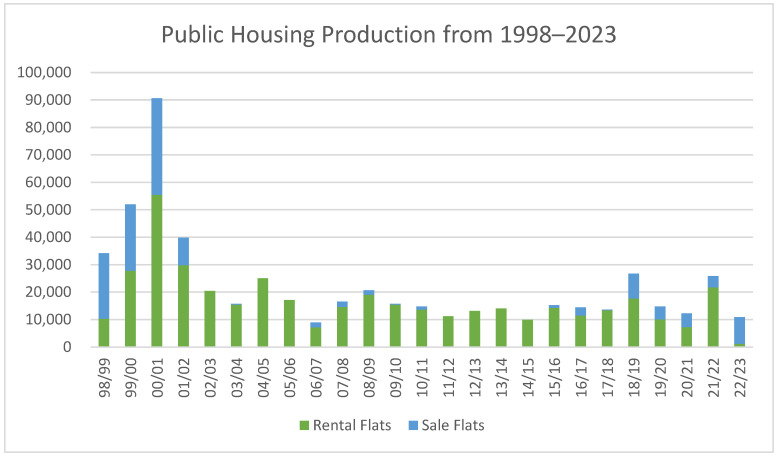
Public housing production from 1999 to 2023. Sources: Housing Authority [73], Housing Authority [74].

**Table 1 ijerph-22-01182-t001:** Average household monthly savings by income quintiles and the types of accommodation 1989/1990–2019/2020 (at the constant price of 2019/20).

Years	Housing Type	Average Monthly Savings by Income Quintiles				
		1st (Lowest)	2nd	3rd	4th	5th (Highest)
1989/1990	PRH tenants	−1095	231	3544	7581	19,012
	Non-PRH tenants	−349	712	−2	5033	23,654
	Home owners	***	***	4070	8260	27,829
1994/1995	PRH tenants	−902	1846	4930	11,711	28,105
	Non-PRH tenants	−707	1149	−1478	1724	30,008
	Home owners	−2466	1550	4346	12,382	33,645
1999/2000	PRH tenants	371	4196	10,551	18,773	34,889
	Non-PRH tenants	57	379	3574	12,606	41,338
	Home owners	−135	4716	8969	18,824	51,330
2004/2005	PRH tenants	−259	4584	9689	18,310	30,976
	Non-PRH tenants	−2350	2099	7109	9372	45,353
	Home owners	−18	4802	10,431	19,326	54,116
2009/2010	PRH tenants	854	4984	11,068	21,209	52,451
	Non-PRH tenants	−3593	−558	5085	11,778	41,929
	Home owners	−891	4895	11,729	22,070	57,672
2014/2015	PRH tenants	2109	8091	14,015	24,182	39,873
	Non-PRH tenants	−2022	1149	4917	13,423	52,910
	Home owners	1155	6447	12,977	24,917	58,615
2019/2020	PRH tenants	2203	7747	16,438	29,494	52,158
	Non-PRH tenants	−1769	1682	6911	17,573	70,587
	Home owners	1120	8991	16,599	31,051	81,813

*** Data suppressed due to small sample sizes; PRH = Public Rental Housing.

**Table 2 ijerph-22-01182-t002:** PRH as an in-kind subsidy to monthly household savings.

	Ordinary Least Square Models on Monthly Household Savings						
	1989/1990	1994/1995	1999/2000	2004/2005	2009/2010	2014/2015	2019/2020
**Household structure**							
Age	30.4 *	50.2 ***	48.4 ***	12.4	19.0	-6.2	13.1
	(3, 57.7)	(22.6, 77.9)	(21.1, 75.7)	(−15.5, 40.4)	(−8.6, 46.7)	(−30.4, 18)	(−16.5, 42.6)
Female	−612.8	−130.2	−833.1 *	−1178.4 ***	−1108.3 ***	−979.1 ***	12.4
	(−1319.4, 93.8)	(−821.7, 561.2)	(−1468.2, −198.1)	(−1805.4, −551.4)	(−1733.9, −482.8)	(−1522.1, −436)	(−625.3, 650.1)
Household sizes	−1262.8	−1644.3 ***	−716.1 ***	−1459.4 ***	−1531.8 ***	−1373.6 ***	−1963.0 ***
	(−1535.5, −990.3)	(−1930.1, −1358.5)	(−987.8, −444.4)	(−1748.9, −1159.8)	(−1831.9, −1231.6)	(−1653.3, −1093.9)	(−2284.2, −1641.2)
Number of children	167.6	102.5	−1174 ***	−1024.9 ***	−1337.7 ***	1383.5 **	−1839.3 ***
	(−245.2, 580.4)	(−315.2, 520.1)	(−1596.7, −751.3)	(−1519.6, −530.3)	(−1869, −806.5)	(1889.3, −877.7)	(−2424.0, −1253.5)
Number of elders	1106.4 ***	803.2 **	513.9	511.5	1234.8 ***	837.5 **	406.4 *
	(479.9, 1732.9)	(197.9, 1408.6)	(−88.3, 1116.1)	(−100.0, 1123.1)	(629.0, 1840.6)	(329.6, 1345.4)	(−146.6, 959.4)
Household monthly income	0.52 ***	0.53 **	0.51 **	0.55 ***	0.61 ***	0.60 ***	0.7 ***
	(0.49, 0.54)	(0.51, 0.55)	(0.50, 0.52)	(0.53, 0.56)	(0.60, 0.63)	(0.59, 0.62)	(0.70, 0.72)
**Types of accommodation (PRH tenants as reference group)**							
Non-PRH tenants	−2302.0 ***	−5343.5 ***	−4482.7 ***	−4785.6 ***	−6959.4 ***	−8698.3 ***	−9186.9 ***
	(−3295.4, −1308.6)	(−6361.7, −4325.4)	(−5419.9, −3545.5)	(−5768.7, −3802.4)	(−8004, −5714.8)	(−9590.7, −7806)	(−10138, −8235.7)
Homeowners	513.4	391.5	1164.5 ***	1668.2 ***	507.8 ***	−42.9	1122.2 **
	(−235.1, 1261.9)	(−320.2, 1103.1)	(501.7, 1827.4)	(977.0, 2359.5)	(−187.9, 1203.5)	(−688.2, 572.4)	(402.4, 1842.1)
Intercept	−4188.4 ***	−3126.8 ***	−4163.1 ***	−814.3	−1616.2 ***	131.6 ***	−2750.6 **
	(−5725, −2651.9)	(−4744.7, −1508.9)	(−5739.7, −2586.5)	(−2452.3, 823.8)	(−3315.1, 82.7)	(−1400.1, 1663.4)	(−4590.0, −911.2)
Model R-squared	39.5%	48.0%	61.7%	69.9%	72.0%	71.6%	83.1%

* *p* < 0.05; ** *p* < 0.01; *** *p* < 0.001.

**Table 3 ijerph-22-01182-t003:** PRH as an in-kind subsidy to monthly household savings after PSM.

	Ordinary Least Square Models on Monthly Household Savings After Propensity Score Matching	
	2009/2010	2014/2015
Household structure		
Age	−41.78 (−221.79, 138.23)	−52.46 (−191.93, 87.01)
Female	1195.92 (−2901.40, 5293.24)	1730.59 (−1106.77, 4567.95)
Household sizes	−77.34 (−176.04, 21.36)	−113.58 ** (−181.30, −45.87)
Dependency ratio	−38.57 (−101.92, 24.78)	−15.45 (−62.93, 32.03)
Household monthly income	0.54 *** (0.48, 0.60)	0.59 *** (0.55, 0.64)
Types of accommodation (PRH tenants as reference group)		
Non-PRH tenants	−9601.83 ** (−16,382.44, −2821.22)	−10,978.07 *** (−15,557.17, −6398.96)
Intercept	655.23 (−12,906.37, 14,216.83)	−599.32 (−10,368.76, 9170.11)
Model R-squared	64%	73%

** *p* < 0.01; *** *p* < 0.001.

## Data Availability

The data are not publicly available due to restrictions by the Census and Statistics Department of the Hong Kong SAR Government.

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
