# Peer review of "Public Housing and Household Savings—A Three-Decade Repeated Cross-Sectional Analysis"

_ijerph, 2025, doi:10.3390/ijerph22081182_

Round 1

Reviewer 1 Report

Comments and Suggestions for Authors

Dear authors,

The study analyzes the impact of public rental housing (PRH) on household savings in Hong Kong, using seven rounds of data from the Household Expenditure Survey, covering the period from 1989/1990 to 2019/2020. The authors analyze how living in PRH affects household savings compared to tenants in private housing, after controlling for income and household structure variables. The study finds that PRH tenants save significantly more resources than private housing tenants, with the gap widening over time.

Positive aspects of the article

The paper addresses an important social issue affecting a significant portion of Hong Kong's population, with approximately 28% of residents living in PRH.

The study adopts an innovative approach to quantifying the value of public housing by examining its impact on household savings, going beyond traditional methods such as imputed rent calculations.

The 30-year period provides valuable insights into how the effects of PRH have evolved over time in Hong Kong.

The conclusions have direct implications for housing policy in Hong Kong, particularly regarding the equitable allocation of limited PRH resources.

The statistical approaches control for important variables, including income and household structure, strengthening the validity of the findings.

Less convincing elements

The methodology acknowledges but does not fully address potential errors in how household expenditures and savings are calculated. For example, expenditures do not include investments or cash gifts to relatives, potentially overestimating savings.

The exclusion of households receiving Comprehensive Social Assistance (approximately 4-5% of the population) may limit the generalizability of findings to all low-income households.

Although the paper identifies associations between PRH and increased savings, it does not fully address potential selection effects or other confounding variables that might explain these differences.

The article focuses heavily on the Hong Kong context, with limited comparison to international housing policies or savings patterns, which could strengthen the theoretical framework.

The article does not explore in depth how PRH benefits might differ across demographic subgroups (for example, by age, education level, or occupation).

Suggestions for improvement

A more comprehensive theoretical framework is needed to link housing policy to savings behavior, drawing on existing literature.

Consider additional statistical approaches to address potential endogeneity or selection bias, such as propensity score matching or instrumental variable approaches.

Some results are presented without sufficient context or explanation. For example, the paper mentions higher savings in 1999/2000 compared to study periods from other years but does not adequately explain this anomaly.

The article sometimes uses "equivalent amount of household savings" and other times "in-kind subsidy" when referring to the value of PRH. Consistent terminology would improve clarity. Define "equivalent amount of household savings" more explicitly in the introduction.

Compare findings more directly with previous studies to contextualize the higher value of savings.

The policy implications section could be strengthened with more concrete, evidence-based recommendations for addressing the identified inequities.

The article addresses an important topic with significant policy implications, uses a novel approach, and analyzes a valuable longitudinal dataset. By strengthening the methodological limitations and improving the elements mentioned above, the paper can be published.

Good luck!

Reviewer 2 Report

Comments and Suggestions for Authors

This paper examines the impact of public rental housing on household savings in Hong Kong, based on seven rounds of the Household Expenditure Survey data (1989/1990 to 2019/2020). For each round of the HES data, ordinary least square (“OLS”) regression models were calculated for the three housing subsets (PRH tenants, non-PRH tenants, and homeowners with no mortgage). However research questions and the case study design is well developed, this paper lacks of a theoretical approach in discussing the aim on the relation between householdincome/ savings and accommodation types such as homeowners/tenants.

Comments on the Quality of English Language

No need.

Reviewer 3 Report

Comments and Suggestions for Authors

This paper examines the impact of PRH on household savings in Hong Kong, based on seven rounds of the Household Expenditure Survey data (1989/1990 to 2019/2020). The manuscript explains the history of housing issues and how the government launched a large-scale housing program (PRH) in the 1970s, providing low-cost rental housing to eligible households. However, the authors do not present an in-depth review of the economies of Hong Kong, mainland China, or other countries such as Singapore and South Korea during this period. Furthermore, they do not discuss global programs and initiatives aimed at alleviating housing challenges for poor and middle-class households in other countries across different timeframes.

My reading indicates that the main problems are: the absence of a convincing model to address all factors influencing household savings, an incomplete literature review, the absence of a comprehensive dataset, a lack of novelty, and a lack of generational debate beyond the case study.

Here you can find the comments in detail:

Abstract

Clearly indicate the significance of the study, the knowledge gap, the research method and analysis approach, and the research contribution.

Introduction

The introduction does not address the issue from a global perspective and lacks details on doping trends and athletes' awareness levels, which would provide a broader context for readers.

The literature review lacks depth and a critical discussion of key components related to doping, its driving factors, and theoretical foundations.

It does not sufficiently review past studies, identify a clear research gap, or provide compelling justification for the research aim and objectives.

Additionally, the manuscript does not discuss relevant theories or models related to athletes’ attitudes toward doping or how external factors, such as lectures or awareness programs, influence these attitudes.

Research Questions

While the study tries to explore how much benefit PRH tenants actually obtain through the PRH program, particularly the value of a PRH unit as an in-kind subsidy compared to non-PRH tenants, the following questions are not well aligned with the research scope:

(1) What were the over-time savings trends of and differences between three groups of households, namely PRH tenants, non-PRH tenants, and private homeowners with no mortgage? (2) How much less did non-PRH tenants saved, compared with PRH tenants and private homeowners? and (3) What were the over-time effects of PRH on HK residents’ household savings?

In general, household savings are multidimensional and depend on factors such as household lifestyle, number of children, educational background, employment sector, and perceived inflation. This study focuses solely on savings, which is not a sufficiently clear index for assessing the impact of an affordable housing policy.

Results and Data Analysis

The analysis presented is relatively basic for a journal like IJERPH. More rigorous data and statistical testing are necessary to ensure that the findings are robust. A clearly structured research model would help address this concern.

The manuscript mentions some limitations, but these are basic components of a well-structured manuscript rather than genuine limitations. Fundamentally, it is problematic for a manuscript with such a large dataset and significant uncertainties to produce qualified findings.

Conclusion

The conclusion is too brief and does not effectively highlight the theoretical and practical contributions of the study. Expanding this section to articulate how the research advances existing literature and its real-world applications would help clarify its significance.

Overall, while the manuscript explores an important topic, addressing the concerns outlined above is essential to enhance its academic rigor and clarity. I encourage the authors to refine the literature review, methodological approach, and analytical depth to ensure a more robust and compelling contribution to the field.

Best,

Round 2

Reviewer 1 Report

Comments and Suggestions for Authors

The authors responded to suggestions for improvement and the article is now well written.

Author Response

Thank you for your comments.

Reviewer 2 Report

Comments and Suggestions for Authors

The renewed version seems to be improved based on the previous comments.

Author Response

Thank you for your comments. 

Reviewer 3 Report

Comments and Suggestions for Authors

Dear Authors, 
Thank you for the opportunity to review the revised version of the manuscript. While I appreciate the authors' effort in addressing the initial feedback, I regret to note that several key concerns raised in my initial review have not been adequately addressed. These unresolved issues significantly impact the academic quality and rigor of the paper.
Thus, I reiterate them here for more consideration:

Abstract
Clearly indicate the significance of the study, the knowledge gap, the research method and analysis approach, and the research contribution. Currently, the abstract lacks clarity in defining these essential components, which are critical for positioning the paper within the academic discourse.

Introduction
The introduction does not address the issue from a global perspective and lacks details on doping trends and athletes' awareness levels, which would provide a broader context for readers. A more comprehensive review of international contexts would help justify the relevance of the study.

Research Questions
While the study attempts to explore how much benefit PRH tenants derive through the PRH program—particularly the value of a PRH unit as an in-kind subsidy compared to non-PRH tenants—the following research questions are not well aligned with the research scope:
(1) What were the over-time savings trends of and differences between three groups of households, namely PRH tenants, non-PRH tenants, and private homeowners with no mortgage?
(2) How much less did non-PRH tenants save, compared with PRH tenants and private homeowners?
(3) What were the over-time effects of PRH on HK residents’ household savings?

In general, household savings are multidimensional and depend on factors such as household lifestyle, number of children, educational background, employment sector, and perceived inflation. This study focuses solely on savings, which is not a sufficiently clear or robust indicator to assess the impact of an affordable housing policy.

Results and Data Analysis
The analysis presented is relatively basic for a journal like IJERPH. More rigorous data treatment and statistical testing are necessary to ensure that the findings are robust and valid. A clearly structured research model would greatly strengthen the analysis.

Although the manuscript mentions some limitations, these are general observations and lack specificity. For a study with such a large dataset and inherent uncertainties, a more critical discussion is expected. The current form raises concerns about the reliability of the conclusions drawn.

Literature Review and Theoretical Background
The literature review lacks depth and critical engagement with relevant research. It does not adequately cover key topics related to doping, its determinants, or theoretical foundations. Furthermore, it fails to define a research gap or justify the need for the current study.

Additionally, while the authors mention that a theoretical framework is provided on pages 3 and 4, no clear framework is actually presented or integrated into the research design. This remains a serious weakness that undermines the conceptual coherence of the paper.

Conclusion
In my initial review, I recommended revising the conclusion to better articulate the theoretical and practical contributions of the study. While the section has been extended, it still lacks clear, meaningful insights on the theoretical implications and empirical relevance of the findings. This results in a missed opportunity to position the study’s contribution to both academic knowledge and policy.

Author Response

Thank you for your reminder. The response memo was now structured in a point-by-point response to reviewers’ comments –

Abstract: Clearly indicate the significance of the study, the knowledge gap, the research method and analysis approach, and the research contribution. Currently, the abstract lacks clarity in defining these essential components, which are critical for positioning the paper within the academic discourse.

1.        We have revised the abstract according to reviewer 3’s comment, particularly addressing the significance of study, current research gap, the research method and analysis approach, and research contribution. We would like to thank reviewer 3 for the comments, as it greatly enhances the readability of the abstract.

Introduction: The introduction does not address the issue from a global perspective and lacks details on doping trends and athletes' awareness levels, which would provide a broader context for readers. A more comprehensive review of international contexts would help justify the relevance of the study.

2.        We recognize the importance of incorporating studies of international contexts to help justify the relevance of the study. Particularly, we would like to draw your attention to the Introductory paragraph and section 1.3 that we conducted literature review on the concept from different policy contexts.

Research Questions: While the study attempts to explore how much benefit PRH tenants derive through the PRH program—particularly the value of a PRH unit as an in-kind subsidy compared to non-PRH tenants—the following research questions are not well aligned with the research scope: (1) What were the over-time savings trends of and differences between three groups of households, namely PRH tenants, non-PRH tenants, and private homeowners with no mortgage? (2) How much less did non-PRH tenants save, compared with PRH tenants and private homeowners? (3) What were the over-time effects of PRH on HK residents’ household savings?

In general, household savings are multidimensional and depend on factors such as household lifestyle, number of children, educational background, employment sector, and perceived inflation. This study focuses solely on savings, which is not a sufficiently clear or robust indicator to assess the impact of an affordable housing policy.

3.        Thank you for your insightful comments. To clarify, the first research question examines the overall savings trends across different groups, providing a comprehensive overview of savings patterns at various cross-sectional levels. The second research question delves deeper by exploring differences in savings estimates among various housing types using OLS models, thereby offering a controlled assessment of the specific benefits that PRH confers in terms of savings. In particular, prior literature estimates the benefits of PRH as low as HKD$240 to up to HKD$3708, while the estimate in our study, while controlling for other household factors that may confound the results, finds a benefit in terms of saving as high as HKD$9186.9. In terms of research and policy implication, this divergence is one that warrants public discussion on the use and return of public resources. The third research question extends this analysis by considering alternative model specifications and accounting for potential confounding variables, further investigating the relationships between savings and household structure. Collectively, these research questions are designed to systematically and thoroughly illuminate the savings advantages associated with PRH. For a more detailed discussion addressing your concerns, please refer to Section 5, where the research questions are explicitly discussed in light of the results. This section provides a nuanced interpretation of the correlations observed in the study.

Results and Data Analysis: The analysis presented is relatively basic for a journal like IJERPH. More rigorous data treatment and statistical testing are necessary to ensure that the findings are robust and valid. A clearly structured research model would greatly strengthen the analysis.

Although the manuscript mentions some limitations, these are general observations and lack specificity. For a study with such a large dataset and inherent uncertainties, a more critical discussion is expected. The current form raises concerns about the reliability of the conclusions drawn.

4.        Thank you for your comments. We acknowledge the importance and the need to conduct more rigorous data treatment and testing. To further validate the robustness of our findings, we conducted additional analyses using a 20% sample drawn from the full dataset, due to data confidentiality and availability constraints imposed by the government. Specifically, we compared PRH tenants with non-PRH tenants by employing propensity score matching (PSM) based on household size, income, gender, age, and dependent ratio to minimize potential confounding effects. We would like to draw your kind attention to section 3, at Table 3 where we present the results, consistent with the main analysis on the part of the association of household monthly income and housing types on household savings. We would like to thank reviewer 3 for suggesting ways to greatly enhance the reliability of this study.

In addition, to provide more critical discussion on the results, especially on the interpretation and implication of how such evidence (that is distinctive from previous imputed estimated benefit of PRH) could inform policy design, we enriched our ideas at section 5.2, 5.3 and 5.4, hoping to bring about a more critical, systematic and comprehensive analysis of impact of the results.

Literature Review and Theoretical Background: The literature review lacks depth and critical engagement with relevant research. It does not adequately cover key topics related to doping, its determinants, or theoretical foundations. Furthermore, it fails to define a research gap or justify the need for the current study.

Additionally, while the authors mention that a theoretical framework is provided on pages 3 and 4, no clear framework is actually presented or integrated into the research design. This remains a serious weakness that undermines the conceptual coherence of the paper.

5.        Thank you for your comments on the literature review and theoretical background. The PRH scheme in Hong Kong is distinctive because it provides eligible households with highly subsidized rental units, granting them de facto lifelong security of tenure and the ability to transfer this right to their descendants. Yet, these households do not possess legal ownership of the property. This hybrid arrangement raises important questions about how such in-kind benefits influence household saving and consumption decisions, particularly compared to conventional models of homeownership or cash assistance. This context creates an important research gap, which is clarified and explicitly listed in paragraph 1.3. Previous studies have established that extensive housing-related policies can profoundly impact household saving rates.  Our study is standing on the ground that housing has a nuanced relationship with savings. However, these studies often focus on homeownership, cash-based subsidies, or some specific changes in housing policy environment, leaving a significant gap in our understanding of the implications of large-scale in-kind housing subsidies—such as Hong Kong’s PRH scheme—on household financial behavior.

Understanding the benefit of PRH as a type of in-kind subsidy is crucial for several reasons. First, it provides empirical evidence on how non-cash, non-ownership-based housing policies affect financial decision-making, household welfare, and intergenerational mobility. Second, with housing affordability and inequality becoming increasingly pressing issues globally, insights from Hong Kong’s experience can inform policy development in other high-density urban settings facing similar challenges. Third, the PRH scheme’s unique features—life-long security, transferability, and absence of legal ownership—allow us to isolate the effects of housing security from the wealth effects typically associated with ownership. This distinction is vital for understanding the true value and behavioral impact of in-kind housing subsidies.

Round 3

Reviewer 3 Report

Comments and Suggestions for Authors

The authors have submitted a revised version of the manuscript.

Based on my understanding, I realized that the second round of revision has made significant improvements. The authors have addressed all previous comments to a reasonable extent. In my assessment, the manuscript now meets the minimum standards and requirements for publication in this journal.
I believe the paper should be considered for publication.